# Human Echinococcosis in the Russian Federation in the 21st Century: A Systematic Review

**DOI:** 10.3390/microorganisms13051122

**Published:** 2025-05-14

**Authors:** Branko Bobić, Tijana Štajner, Vladimir Ćirković, Jelena Srbljanović, Olivera Lijeskić, Neda Bauman, Đorđe Zlatković

**Affiliations:** Group for Microbiology and Parasitology, Center of Excellence for Food and Vector-borne Zoonoses, Institute for Medical Research, National Institute of Republic of Serbia, University of Belgrade, 11129 Belgrade, Serbia; tijana.stajner@imi.bg.ac.rs (T.Š.); vladimir.cirkovic@imi.bg.ac.rs (V.Ć.); jelena.srbljanovic@imi.bg.ac.rs (J.S.); olivera.lijeskic@imi.bg.ac.rs (O.L.); neda.bauman@imi.bg.ac.rs (N.B.); djordje.zlatkovic@imi.bg.ac.rs (Đ.Z.)

**Keywords:** human echinococcosis, Russian Federation, 21st century, epidemiology

## Abstract

According to the WHO, echinococcosis is a neglected tropical disease of global importance. The Russian Federation (R.F.) is traditionally considered an endemic area of echinococcosis. This study aims to analyze the state of human infection in the R.F. in the 2000–2021 period, for which there is not enough consolidated data. Epidemiological data on human echinococcosis in the R.F. from 1 January 2000 to 31 December 2021 were collected through literature research (both published and grey literature) and official reports. From the 2022 selected records, 12 full-text articles, three doctoral dissertations, and 17 official reports were analyzed, all of which met the criteria for inclusion in the study. In the R.F., in the period from 2000 to 2021, echinococcosis (cystic (C.E.) and alveolar (A.E.) echinococcosis) has been continuously registered in humans (0.4–0.22 cases/100,000 inhabitants). Until 2013, the incidence of echinococcosis did not change (Pearson’s r (N = 13) = 0.288, *p* = 0.340), but in the period 2013–2021 it decreased significantly (Pearson’s r (N = 9) = −0.709, *p* = 0.032). In that period, the incidence of C.E. decreased significantly (Pearson’s r (N = 9) = −0.717, *p* = 0.035), while the incidence of A.E. did not change (Pearson’s r (N = 9) = −0.518, *p* = 0.154). The infection is registered annually in 30 out of 86 federal units. The C.E. infection rate was significantly higher in the European part (0.46/100,000 population) (2 = 33,783. r < 0.00001) than in the Asian part of the R.F. (0.32/100,000 population), where A.E. was more widespread. Within the European part, the frequency of C.E. infection was significantly higher in the southern (0.70/100,000 inhabitants) (χ^2^ = 806.67, *p* < 00001) than in the central and northern parts (0.25/100,000 inhabitants). The incidence of C.E. per federal district was positively correlated with rural population representation (Pearson’s r (N = 8) = 0.866, *p* = 0.005). Every year, although in small numbers, deaths caused by echinococcosis (in the period 2009–2020—66 deaths) were registered in the R.F., significantly more often caused by A.E. than C.E. (χ^2^ = 39.4401, *p* < 0.00001). Our results indicate that, between 2000 and 2021, echinococcosis was still generally endemic in the R.F. The incidence of C.E. has demonstrated a decreasing trend, especially after 2014, while the rate of A.E. remained unchanged.

## 1. Introduction

Echinococcosis is a worldwide zoonosis caused by cestodes of the genus Echinococcus sp. of the family *Taeniidae*. In Europe, zoonosis is most often caused by *Echinococcus granulosus sensu lato* (E.G.), a globally widespread parasite and the causative agent of cystic echinococcosis (C.E.), and *Echinococcus multilocularis* (E.M.), the causative agent of alveolar echinococcosis (A.E.), widespread in the northern hemisphere [1].

The parasite’s life cycle includes development in a permanent (carnivorous) and intermediate host (herbivores and omnivores). In Europe, the leading definitive hosts for E.G. are dogs, while for A.E. they are foxes. Infection of the permanent host occurs after the consumption of organs of infected intermediate hosts. The adult cestode inhabits the small intestine of the final host, and the segments of the cestode (proglottids) containing eggs, or free eggs, are excreted by feces into the environment. The consumption of food or water contaminated with helminth eggs is a route of infection in intermediate hosts [2]. The course of infection in the intermediate host’s internal organs, most often in the liver, lungs, and CNS, implies the development of metacestodes that subsequently form cysts.

As intermediate hosts, humans are also infected by accidental ingestion of eggs through contaminated unwashed raw fruits or vegetables, especially those growing near the ground, contaminated water, or unwashed hands. Humans have no role in spreading parasites, but infection is a serious medical problem. It is classified as a neglected tropical disease, condition, or syndrome of global health importance, which prevention and control are supported by WHO [3].

Human echinococcosis is a chronic disease. The more widespread C.E. is a disease with a low mortality rate, while A.E. is one of the most important fatal zoonotic helminth infections in the Northern Hemisphere [4]. Mortality from C.E. is 2 to 4%, but in cases of late diagnosis, it can increase significantly. Without timely diagnosis and adequate treatment, mortality from A.E. is over 90% within 10–15 years [1].

The incidence of infection in the world, as well as in Europe, varies from areas with sporadic cases to highly endemic [2,4,5]. The territory of the former Soviet Union is considered an endemic area for C.E. and A.E. infection [1]. Among the former republics of the Soviet Union, the largest number of cases is registered in Moldova, the republics of the South Caucasus, Kazakhstan, Kyrgyzstan, Uzbekistan, and Turkmenistan [6]. The Russian Federation (R.F.) is also considered an endemic area for both C.E. and A.E. [7], although data on its prevalence are scarce. Negative trends in the economic life of the Russian Federation during the 1990s led to dysfunction in all parts of society, including the areas of health and veterinary services and prevention. After 2000, the beginning of economic recovery (according to World Bank GDP data, https://data.worldbank.org/indicator/NY.GDP.PCAP.CD?locations=RU (accessed on 1 May 2021) in the R.F. was a prerequisite for the recovery of these services in these new circumstances. Recent studies on the incidence of infection in Europe or Asia do not cover the R.F. [8,9,10] and although it is evident that changes in socio-economic relations in the last twenty years in Russia have significantly influenced the spread of other tapeworm infections [11], this study therefore aims to review current trends in the R.F. in the 21st century.

## 2. Materials and Methods

### 2.1. Search Strategy

In accordance with the PRISMA guidelines [12], we conducted a systematic database search of the published literature and local information sources on the epidemiology (incidence, prevalence, incidence, age, sex, and geographical distribution) of human echinococcosis in the R.F. (Appendix A). The research period was from 2001, as data for one year is published starting from the following year, until 2023 in order to collect as much data as possible, especially on the most recent years covered by the research.

### 2.2. Databases and Other Sources

Both international and Russian databases of published and grey literature (master’s and doctoral theses) were searched from 2022 to 2023. For published data, two international databases (PubMed—http://www.ncbi.nlm.nih.gov/pubmed; and Google Scholar—https://scholar.google.com/scholar) and three Russian databases (DVGMU Library, Far Eastern Library of the Medical State University—www.fesmu.ru/elib/Search.aspx?Catalogue=1; eLIBRARI.RU, Scientific Electronic Library—https://elibrari.ru/; and Ciberleninka—https://cyberleninka.ru/) were searched. In addition, the international databases OpenGrey (http://www.opengrey.eu/) and Russian Scientific Library, Library of Dissertations (http://freereferats.ru/index.php?cat=91&page=14) were searched. The following search terms were used: Echinococcus OR granulosus OR alveolar OR hydatidosis AND Russian Federation. For Russian databases, the same search terms were used but in Russian only. Furthermore, the official websites of Russian government agencies were searched for official reports and overviews as well as epidemiological bulletins, using the same search terms in Russian.

### 2.3. Selection Criteria

Search results were collated and re-checked for publication date (2001–2023), and duplicates and insufficient records were removed. The titles and abstracts were then screened for relevance using the following exclusion criteria: (1) studies concerning other parasites; (2) studies conducted outside the R.F.; (3) studies that did not include data on human infections; (4) studies that did not include data for the study period; (5) studies presenting data not related to the epidemiological features of echinococcal infection, but rather, focusing on clinical features, therapy, or biology of the parasite; (6) studies containing a general overview of the topic without original data; and (7) studies presenting duplicate data. The same exclusion criteria applied to the full text; an additional exclusion criterion was that they merely repeated epidemiological data published in official reports. Official reports were considered acceptable data sources.

### 2.4. Data Extraction and Generation

The incidence data provided in the selected literature and reports were used directly. In cases where the number of echinococcosis cases was reported, the incidence was calculated using official census data on population size. Demographic data, data on population density, rural population distribution, and GDP per capita/year by federal districts in the R.F. were taken from the official statistical publications of the State Statistics Service–Rosstat (all available online). The statistical analysis was carried out using the univariate analysis of variance, Pearson’s two-sided correlation, and the chi-square test (χ^2^) or the Fisher’s exact test.

## 3. Results

The database search results are shown in the flow diagram (Appendix A). A total of 54 relevant entries were identified and included in the review; of these, 10 were published papers, three were doctoral theses, and 39 were official annual reports of the R.F. ministries. All relevant references are included in the analysis.

### 3.1. Incidence of Echinococcosis in RF

According to official federal data, a total of 11,272 cases of echinococcosis were registered in the R.F. over 22 years (2000–2021) [13,14]. During this period, the incidence ranged between 0.16 and 0.41/per 100,000 population, with most cases reported in 2012 and the least in 2020 (Figure 1). Until 2013, C.E. and A.E. cases were not reported separately, so only the total number of patients with echinococcosis was available.

Most cases (88.20–90.72%) were diagnosed in patients who sought medical attention for pre-existing symptoms. The diagnosis of echinococcosis in reported cases is based on ultrasound, radiography, computed tomography, or surgery [13].

Some of the reported cases were imported. Of the 76 patients with echinococcosis registered in St. Petersburg between 2000 and 2011, 30 (39%) were foreign nationals [15]. A similar situation was recorded in Moscow between 2011 and 2013: out of 153 (23) registered cases, 35 were detected in foreigners [16]. On the other hand, studies in some administrative regions that analyzed clinical records recorded a higher number of cases than officially reported. Thus, an analysis of clinical records in the Orenburg region from 1994–2012 found 1393 diagnosed cases of echinococcosis, while 1186 cases were reported in federal reports in the same period [17].

Overall, in the R.F., the infection rate decreased significantly over the entire period from 2000 to 2021 (Pearson’s r (N = 22) = −0.495, *p* = 0.019). From 2000 to 2012, there was no significant change (Pearson’s r (N = 13) = 0.288, *p* = 0.340), but from 2013 to 2021 the infection rate decreased significantly (Pearson’s r (N = 9) = −0.723, *p* = 0.028). In some R.F. units, however, the trend was different and also changed over the analyzed period. In Dagestan, between 2001 and 2005, there was a significant increase in the incidence of C.E. (1.2, 2.0, 2.1, 2.4, 2.8/per 100,000 population retrospectively) (Pearson’s r (N = 5) = 0.970 *p* = 0.006) [18]. The number of cases in Astrakhan increased continuously between 2001 and 2015 from 21 to 70/per 100,000 population and then decreased to 45/per 100,000 population from 2016 to 2021 [19]. The trend was different even in the same federal district between individual units [20].

Data on the distribution of reported cases by federal units of the R.F. are available for the fourteenth annual period 2008–2021 [21]. The R.F. is administratively divided into 86 federal units, and between 2008 and 2021, echinococcosis was unreported only in the Republic of Ingushetia. During that period, the number of federal units in which infections were reported varied annually from 51 to 84 (mean 63.64 ± 7.31) (Figure 2), but no trend was detected (Pearson’s correlation r (N = 14) = 0.357, *p* = 0.211). Over the fourteen-year period from 2008 to 2021, infection was registered in 29.5% of federal units each year, while in 8.4% of units it was registered less often than every other year (Figure 3).

However, the number of reported cases by the federal unit does not fully reflect the true picture. In Saint Petersburg, out of 46 cases among Russian citizens registered from 2000 to 2011, 21 (46%*) came from other regions of the R.F. [15], and in the Moscow regionok 13/25 (52%) in 2011 or 29/47 (62%) in 2017 [16].

The Federal units of the R.F. are organized into eight federal districts. The cumulative frequency of echinococcosis in the federal districts of the R.F. for the period of 2008–2020 is presented in Figure 4. The analysis showed that the infection rate varied significantly between federal districts (F (1,8) = 16.026), *p* = 0.005). In general, the infection rate is significantly higher (χ^2^ = 33.783. *p* < 0.00001) in the European (0.46/100,000 population) than in the Asian part (0.32/100,000 population) of the R.F. Within the European part, the frequency of infection is significantly higher (χ^2^ = 806.67 *p* < 00001) in the southern (0.70/100,000 inhabitants) than in the central and northern districts (0.25/100,000 inhabitants).

Differences in infection rates between federal districts of the R.F. were not correlated with GDP per capita (Pearson’s correlation r (N = 8) = −0.650, *p* = 0.081) or population density (population/km^2^) (Pearson’s correlation r (N = 8) = −0.171, *p* = 0.686). The infection rate is positively correlated with the rural population by federal districts (Pearson’s correlation r (N = 8) = 0.866, *p* = 0.005). Official data on GDP value, population density and the share of rural population in federal districts for 2013 were used for the analysis [22].

Infection has been recorded in all age groups, from children aged 3 to people over 80 (Rospotrebnadzor 2008–2021). The frequency of infections in children up to 17 years of age in the period from 2008 to 2021 (Gr 5) decreased significantly (Pearson’s correlation r (N = 14) = 0.912, *p* = 0.000) [14].

A small number of echinococcosis deaths were registered in the R.F. every year. From 2009 to 2021, 66 deaths were registered in total (between 2 and 11 cases per year) [14].

As of 2014, cases of C.E. and A.E. are reported separately in R.F., so it was possible to analyze the data for each infection individually.

### 3.2. Cystic Echinococcosis

Between 2013 and 2021, 3784 cases of C.E. were registered in the R.F. [14], and the incidence decreased significantly during that period (Pearson’s correlation r (N = 9) = −0.717, *p* = 0.030) (Figure 1). In that period, the incidence of infections among children up to 17 years of age decreased significantly (Pearson’s correlation r (N = 8) = −0.729, *p* = 0.040) (Figure 5).

C.E. is more often diagnosed in women (59% in 2012–54.4% in 2019) [23,24]. However, this is not the case in all parts of the R.F. Although the serological examination for echinococcosis among the population of the Far Eastern Federal District in the period 2009–2017 showed a higher prevalence in women (7.04 ± 0.48%) than in men (5.5 ± 0.51%), in the territory of Khabarovsk there was no difference (5.6 ± 0.53/5.8 ± 0.66), and moreover, in the Jewish Autonomous Region the prevalence was higher in men (24.7 ± 2.94/38, 6 ± 7.34) [20]. According to the data obtained in clinical studies in 33 administrative units of the R.F. during 2019, echinococcosis was also more often diagnosed in women (54.4%) but not in all units (44.4% in the Tymensk region, 43.7% in the Krasnoyarsk region) [24]. The same study showed that gender distribution also depended on age; men dominated in the 15–39 age group (61%), while in the over 60 group women led (70%). This difference was also found in Stavropol (North Caucasus Federal District) [25]. In the surgical clinic in Samara (the sixth largest city in the R.F.), 56% of patients treated between 2002 and 2018 were women, but of note, women do comprise 56% of the total population of this area (60% in 2002 to 2010 and 53% in 2010 to 2018) [26]. Furthermore, in Sechenov-Moscow University Hospital, 66% of cases treated between 1996 and 2017 were men [27].

Infection is traditionally more common in rural areas of the R.F. than in urban areas. Between 2001 and 2005, infections were registered four times more often in rural areas (0.7–0.8/0.2 per 100,000 inhabitants) [18]. In the Far Eastern Federal District, however, there were no differences in the seroprevalence of C.E. infection between urban (5.8 ± 0.42%) and rural populations (5.8 ± 0.41%) between 2009 and 2017. (20). According to data from 39 federal units in 2019, an equal number of patients were detected in the rural and urban populations [24]. In the period 2013–2020, 25 deaths from cystic echinococcal infection were registered in the R.F. (Figure 6).

In general, it was established that the main factors responsible for the persistence of C.E. in the R.F. were non-compliance with the regulations relevant for controlling the slaughter of domestic animals and the disposal of the organs of infected animals and a lack of timely deworming of dogs [13]. Case-control studies in the Orenburg region from 1994 to 2012 showed that C.E. occurs significantly more often in dog breeders who raise domestic animals than in people who have no contact with dogs (1.7/0.4 cases per 100,000 population). The risk of infection in households increases when dogs are not dewormed (93.9% in households with registered patients/84.2% of households without registered patients ) (OR = 2.9), when there is no veterinary supervision during slaughter of livestock (in 93.9% in households with registered patients/78.2% of households without registered patients) (OR = 4.3), and when dogs are fed with fresh entrails (in 96.8% in families with registered patients/81.7% of households without registered patients) (OR = 6.7) [17]. According to epidemiological studies in the Astrakhan region in the period between 2000 and 2021, 89.7% of infected people did not wash their hands regularly before eating or after returning home, 9.7% ate unwashed fruits and vegetables, and 0.6% had close contact with stray dogs [19]. Data for the highly urbanized area of St. Petersburg between 2000 and 2011 indicate the importance of continuous contact with dogs as a risk factor, confirmed by 52.1% of patients [15]. However, the possibility of contamination via water cannot be ruled out as a potential risk factor for the infection either, because the results of analyses conducted in 2015 revealed the contamination of water with pathogenic parasites in 0.1% of samples from central water supply systems, in 1.2% of soil samples in sanitary protection zones of springs, and in 1.0 % of surface waters in R.F. [13].

### 3.3. Alveolar Echinococcosis

From 2014, when A.E. was registered as a separate entity, until 2021, 420 cases of A.E. were registered in the R.F. [14]. Before 2014, 259 cases of A.E. were reported from 2007 to 2012 [23], and 47 in 2013 [13]. In the period from 2013 to 2021, the infection rate ranged from 0.01 to 0.05 and did not change significantly (Pearson’s correlation r (N = 9) = −0.518, *p* = 0.154).

Not only is A.E. registered less often than C.E. in the R.F. (714 C.E./467 A.E. between 2013 and 2021), but also in fewer federal units. In the period 2013–2015, cases of A.E. were registered in 31 and C.E. in 80 federal units (χ^2^ = 63.802, *p* < 0.0001) [13]. However, the number of administrative units where A.E. was registered increased from 15 in 2003 to 31 from 2013 to 2015 [13]. A.E. is diagnosed more often in the Asian federal units (Sakha, Krasnoyarsk, Ata, and Khabarovsk, Tomsk, Omsk, Irkutsk, Chelyabinsk, Magadansk, Chukotsk) than in the European ones (Perm, Rostov, Samara) [28]. Cases were also registered in Moscow, but all of them were imported. Thus, of all registered cases of A.E. in Moscow in 2013, 7 originated from other regions of the R.F. (two cases from the Smolensk Region and the Republic of Kabardino-Balkaria, and one case each from the Kaluga Region, the Krasnoyarsk Territory, and Armenia), while there was no data on one case [16]. Among 224 cases of A.E. registered between 2010 and 2015, 3.1% were diagnosed in children up to 17 years old, 57% in women, and 66% overall in the rural population [13,23]. Between 2013 and 2020, 18 cases of infection with fatal outcome were officially recorded in the R.F. (Figure 7) [14]. The cases were registered in a smaller number of administrative units. Thus, from 2014 to 2021, nine deaths were recorded in five administrative units (Altai Krai, Krasnoyarsk Oblast, Kaliningrad Oblast, Sverdlovsk Oblast, Kurgan Oblast). The mortality rate was significantly higher in patients with A.E. than in those with C.E. (χ^2^ = 39.4401, *p* < 0.00001).

A.E. is traditionally detected in natural foci, but mixed foci are also registered today. For example, the formation of mixed foci in the Omsk region confirms a higher percentage of seropositivity among urban residents (9.7%) than among rural residents (1.7%) [13]. However, it is still considered that people exposed to higher risk are those who, due to the nature of their work, stay in nature (forestry workers, collectors and buyers of mushrooms and berries), come into contact with the carcasses of wild animals (fur workshop workers) or hunters, consequently, preventive measures are primarily aimed at them [29].

## 4. Discussion

In the 1990s, the rate of echinococcosis in the R.F. increased from 0.1 in 1991 to 0.3/per 1,000,000 inhabitants in 2001 [13]. According to the data collected and analyzed during this systematic review, the infection rate generally declined after 2000. A significant decline was noted after 2013, due to a decrease in the incidence of C.E., while the incidence of A.C. did not change, which is consistent with Ermakova’s findings for the 2017–2021 period [30]. It can be assumed that the incidence of C.E. also decreased in the period 2000–2013, but not enough to lead to a significant decrease in the overall rate of echinococcosis.

However, data on infection rates based on official reports should be addressed with caution. The first reason for this lies in the non-reporting of all cases to the central public health service, as other authors point out [7], which is also illustrated by the smaller number of reported cases than the actual number of C.E. cases according to hospital records in certain areas of R.F. [17]. Irregular reporting of cases is a common problem in other European countries as well [8]. Since the infection in humans can be asymptomatic, and since most of the reported cases were diagnosed after the patients registered for medical examination due to the presence of symptoms (88.2–90.72%), it could be assumed that this may also be the cause of the inaccuracy of the incidence data. The results of seroepidemiological studies conducted in certain areas of the R.F. indicate a significantly higher incidence of C.E. than the one based on case reports. According to official estimates, the seroprevalence of *E. multilocularis* infection is three times higher than the prevalence based on reported cases [31]. If we consider that according to Frideret et al. [32], the prevalence of liver echinococcosis based on abdominal ultrasound screening is two to three times higher than the prevalence determined by serological screening, it becomes clear that a significant number of cases remain undiagnosed/underreported in the R.F.

The asymptomatic period of C.E. (5- over 20 years) [13] is of particular importance for the analysis of the incidence trend. Accordingly, a significant number of cases reported in the 2000–2013 period represented patients that were probably infected in the last decade of the 20th century, and a portion of the infections acquired in the 2014–2021 period have yet to be diagnosed. However, we believe that the officially recorded decline in C.E. infection rates in the 21st century corresponds to the real picture because there is no reason to believe that the diagnosis/reporting system functioned more irregularly in the later years of the study period. Based on the decline in the rate of C.E. among children under the age of 15 in the 2008–2021 period, we should expect a further decrease in the rate of infection in the following years as well.

The same factors that make it difficult to get a more accurate picture of the frequency of C.E. also affect A.E. data. However, we assume, due to more severe clinical manifestations and shorter asymptomatic periods, that the number of reported cases is more realistic, as well as the invariance of the incidence.

C.E. is spread over the entire territory of the R.F. The only exception, probably due to underreporting of cases, is the territory of the Republic of Ingushetia, unit of the South Caucasus District, the area with the highest incidence of C.E. in the R.F.

Although the R.F. is generally considered an endemic area, there are significant differences in the frequency of reported cases between administrative regions. Between 2008 and 2021, C.E. was registered in 30% of federal units each year and sporadically in 13% (in seven or fewer years over the 14-year period). This is likely the result of the actual epidemiological situation rather than of inconsistent reporting, as there is no reason to assume that disease reporting malfunctions are present in specific years only. In addition to the degree of exposure to infection, other reasons lead to differences in the distribution of reported cases by administrative units. The number of reported cases depends on the stage of socio-economic development of individual areas. In urban areas, such as Moscow and St. Petersburg, a significant number of cases are reported among people who moved to the city for professional or economic reasons. The data is also influenced by the availability and equipment of health services, especially surgery. For example, only one case of echinococcosis was officially reported on the territory of the Republic of Chechnya between 2010 and 2013, but residents of Chechnya were operated on in Rostov-on-Don, in the neighboring federal unit, Rostov Region, in 2005, 2010, 2011, and 2015 [33]. A clearer insight is provided by the analysis of the frequency of infection by federal districts, which include several units (86 federal units are grouped into 7 districts). Interestingly, during the analyzed period, the incidence of C.E. was higher in the European than in the Asian part of the R.F., while this ratio was reversed in the early 1980s [34]. In the European part, the incidence is significantly lower in the central and northern regions than in the southern regions (regions with developed extensive animal husbandry in which dogs play a significant role in disease spreading). The area with the highest prevalence is the North Caucasus Federal District, which is confirmed by other authors [35].

Between 2008 and 2021, the frequency of echinococcosis in the federal districts was directly proportional to the representation of the rural population but not to GDP per capita or population density (number of people/km). As echinococcosis is a consequence of contact with a contaminated environment, it is logical that it is correlated with the representation of the more exposed rural population. In contrast, the incidence of *Taenia solium* infection in the R.F. correlates negatively with GDP per capita due to the consumption of cheaper uncontrolled pork from informal markets [11].

C.E. is traditionally an infection of rural areas of the R.F., as well as throughout Europe [8]. From 2001 to 2005, the infection was registered four times more often in rural areas than in urban areas. Data on the equal number of diagnosed cases among the rural and urban populations in 33 federal subjects from 2019 [24] seem contradictory, but in 2019, 86% of the R.F. population lived in urban areas [36], so the infection is more widespread in rural areas. Moreover, the North Caucasus Federal District, the district with the highest infection rate, is the only district in the R.F. where more than 50% of the population lives in rural areas [36], and extensive livestock farming, including numerous guard dogs, is widespread. Studies of EC seroprevalence in the Far Eastern District also did not show a difference between urban and rural populations. Given that this is an area in the far northeast of the Russian Federation with extreme climatic conditions, although there is no specific data in the literature to explain this, it is clear that the conditions for the spread of infection are specific, e.g., in terms of the possibility and frequency of contact with a contaminated environment, preservation of the vitality of parasite eggs, the number of stray dogs, and the conditions for their survival, etc.

A.E. is also more common in rural areas; people who live or stay in natural hot spots are still most exposed to infection. However, the data confirm the formation of hotspots in urban areas due to increased populations of small rodents and foxes and their appearance in cities where it is easier to find food. From 1993 to 2015, the number of foxes in the R.F. increased by 2.6 times [13]. Therefore, it can be assumed that the proportion of urban patients will also increase. The number of areas in which cases are registered is also increasing. Although the infection is spreading mainly in the Asian part, cases are also registered in some European units of the R.F.—in Rostov and Samara. A study on the dynamics of registration of *E. multilocularis* infection in wild animals in the Central Federal District of Russia (units Vladimir, Ryazan, Moscow, Tver, Orilsk, Bryansk) and the Republic of Karelia from 2007 to 2018 showed the existence of natural foci of infection in the Vladimir and Ryazan regions [37].

According to official data in the R.F., most cases of echinococcosis have been reported in women (59% in 2012–54.4% in 2019) [23,24], but not in all areas. As most of the population in the R.F. is female (53–54% in the 2000–2022 period) (number of men and women) [38], it is clear that there is no significant difference in the distribution of infection by gender. The best example is the data on C.E. from Samara [26]. Echinococcosis occurs in all age groups, and the gender distribution of C.E. has been observed to vary between age groups [24]. Due to the length of the asymptomatic period, especially in C.E., it is not possible to gain a reliable insight into the age distribution of the infection.

Estimates of the global burden average 285,500 disability-adjusted life years (DALYs) for human C.E. and 666,434 DALYs for AE (1). Consistent with the more severe clinical presentation and uncertain prognosis, a significantly higher mortality rate from A.E. than from C.E. has been demonstrated in the Russian Federation.

Future trends on the incidence of echinococcosis in the R.F. depend on the implementation of preventive measures. Although few studies have been conducted on risk factors for the development of echinococcosis, they provide basic guidelines for their definition. However, the peculiarities of living conditions in certain parts of such a large country require further research. A prevention program was introduced in 1983 [34] and is updated at regular intervals (1996, 2003, 2014). The measures adopted in 2014 [39], with minimal changes compared to previous regulations, in addition to general measures (epidemiological monitoring, health education, and mandatory supervision of animal slaughter), foresee special measures for the suppression of echinococcosis. As part of these special measures, regular health checks are prescribed for persons who are professionally exposed to the risk of echinococcosis (hunters, shepherds, reindeer herders, herders, employees on fur farms, livestock farms, zoos, workers in fur workshops, veterinarians, people engaged in breeding dogs, forestry workers, mushroom or berry pickers and buyers) as well as their family members [29]. The same regulations stipulate that stray dogs are captured and kept in animal shelters where deworming is carried out. A deworming procedure conducted 5–10 days before their release into nature is also mandatory for working, draft, and hunting dogs. Cement surfaces intended for defecation with metal containers for collection and chemical disinfection of feces are also planned. In the settlements, special places marked with signs for walking pets and containers for collecting feces must be set up. A monthly sanitary and helminthological inspection of places where pet dogs are kept or bred (farms and households) is planned. In order to prevent epizootic diseases, each settlement and hunting lodge must have special rooms for stripping and initial processing of animal skins, which must provide running water and have smooth surface floors, walls, and equipment. Food consumption is strictly prohibited on these premises. Waste from leather processing and infected carcasses are burned. The application of these measures has certainly contributed to the reduction in the incidence of C.E. in the R.F. However, the full effect depends on the consistency and completeness of their application, which are questionable. The main problems are still the non-enforcement of regulations in the slaughter of domestic animals, the implementation of disinfection, and the lack of effective measures to regulate the number of dogs and their deworming [40]. Fresh and thorough insight into success rates of specific measures, foreseen and undertaken to overcome these prevention challenges, can only be obtained by further monitoring of the incidence of echinococcosis in the R.F.

## Figures and Tables

**Figure 1 microorganisms-13-01122-f001:**
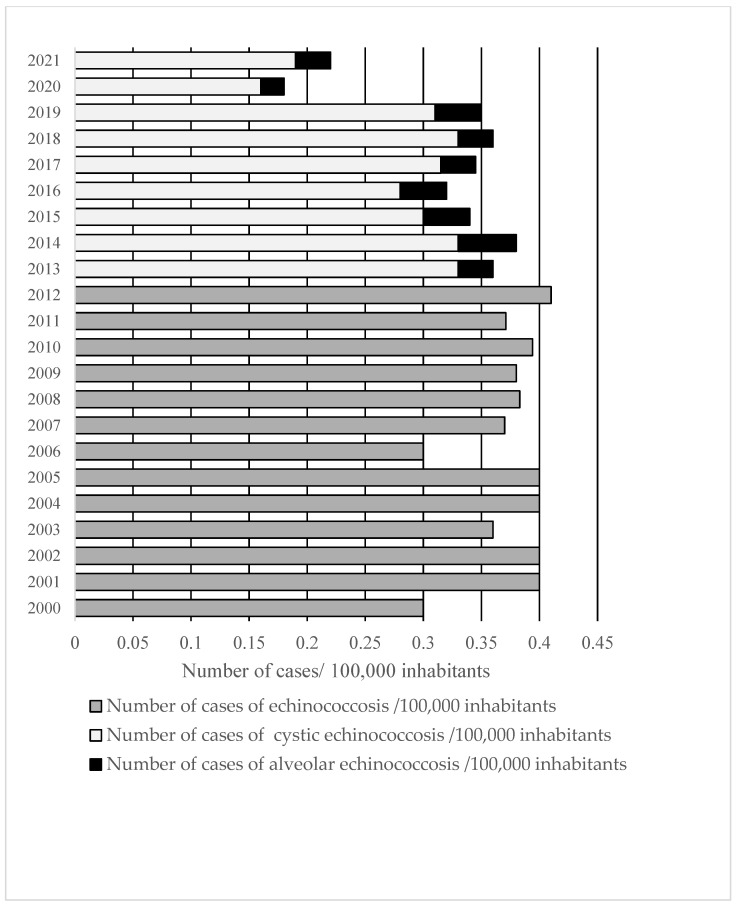
Number of cases of echinococcosis per 100,000 inhabitants officially reported in the R.F. (2000–2021).

**Figure 2 microorganisms-13-01122-f002:**
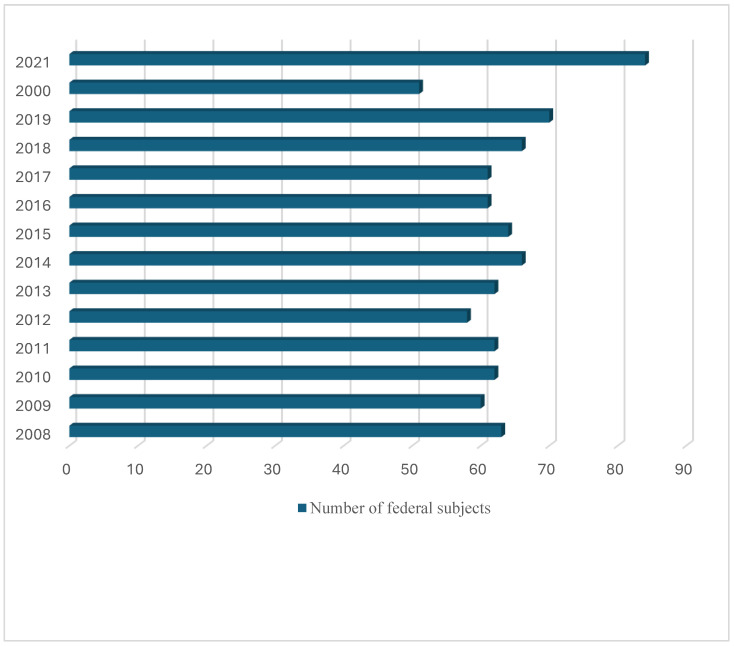
Number of federal units of the R.F. in which the echinococcosis was reported between 2008 and 2021.

**Figure 3 microorganisms-13-01122-f003:**
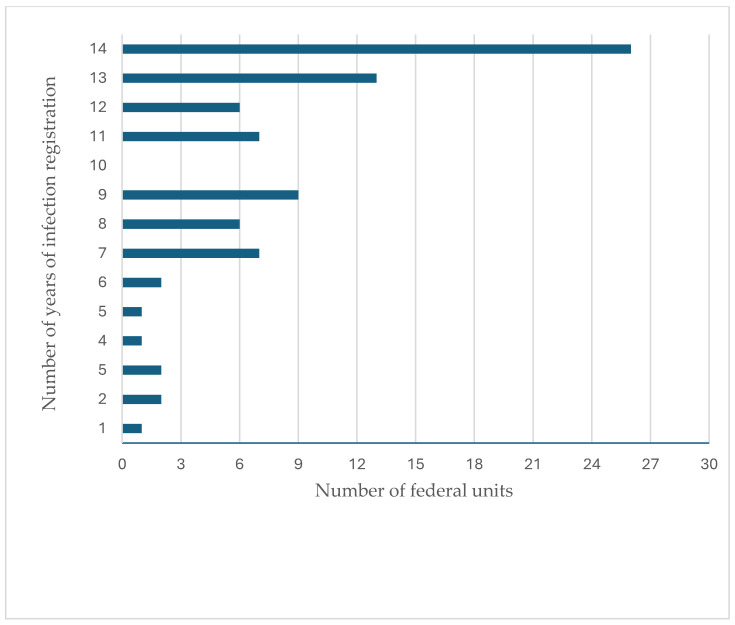
Distribution of years of echinococcosis registration according to the number of federal units during the 2008–2021 period.

**Figure 4 microorganisms-13-01122-f004:**
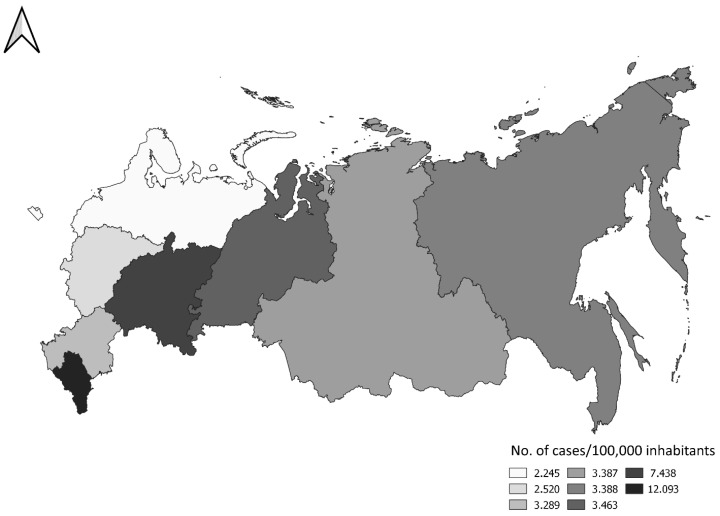
Distribution of officially reported cases of echinococcosis per 100,000 inhabitants by federal units of the R.F. (2008–2020).

**Figure 5 microorganisms-13-01122-f005:**
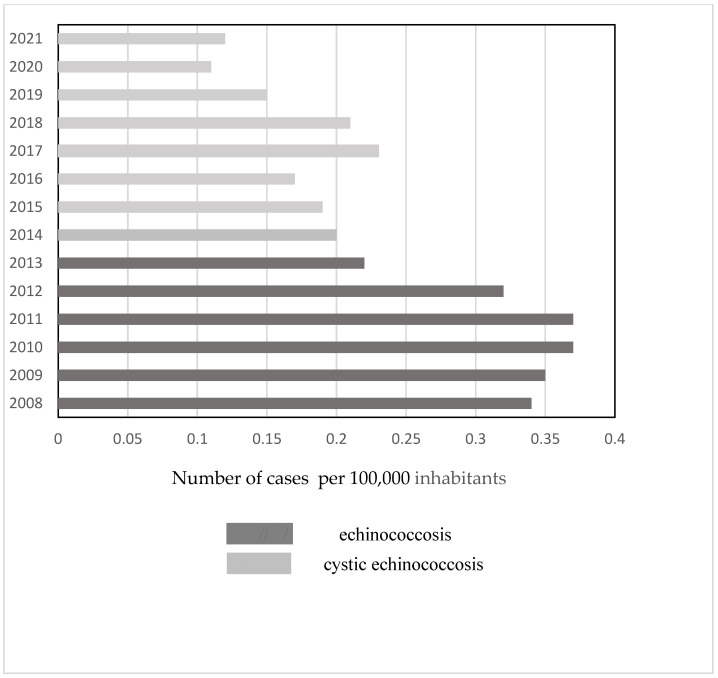
Prevalence of echinococcosis in children under 17 years of age reported in R.F. (2008–2021) per 100,000 inhabitants.

**Figure 6 microorganisms-13-01122-f006:**
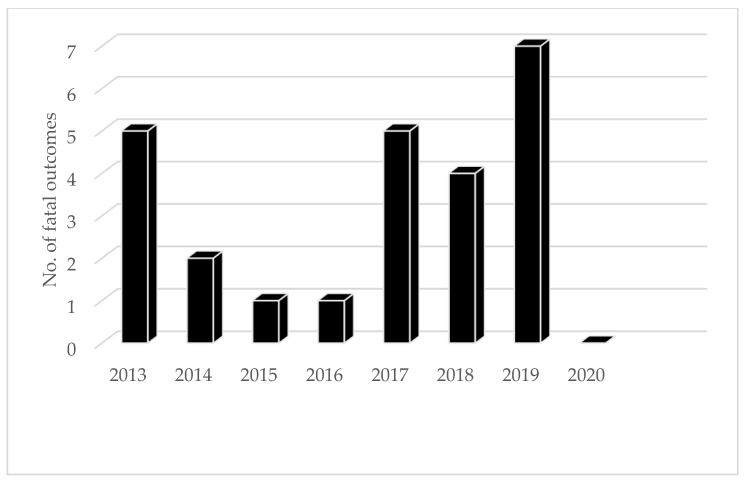
Number of fatal outcomes from cystic echinococcosis in the period 2013–2021 in R.F.

**Figure 7 microorganisms-13-01122-f007:**
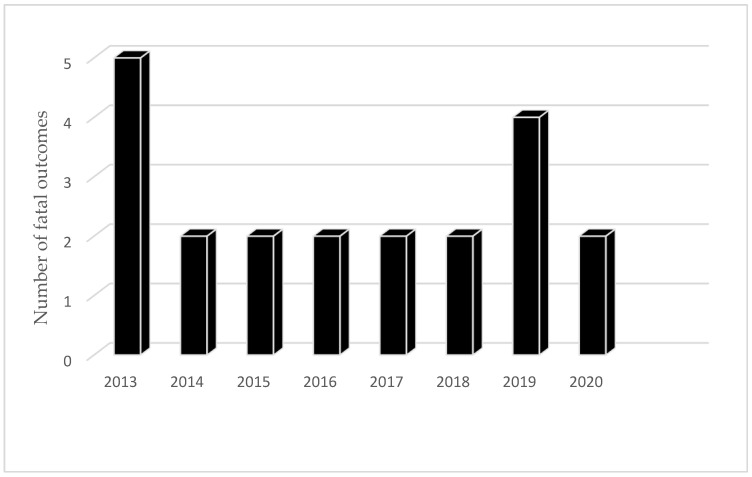
Number of fatal outcomes from alveolar echinococcosis in the period 2013–2021 in R.F.

## Data Availability

For published data, two international databases (PubMed—http://www.ncbi.nlm.nih.gov/pubmed; and Google Scholar—https://scholar.google.com/scholar) and three Russian databases (DVGMU Library, Far Eastern Library of the Medical State University—www.fesmu.ru/elib/Search.aspx?Catalogue=1; eLIBRARI.RU, Scientific Electronic Library—https://elibrari.ru/; and Ciberleninka—https://cyberleninka.ru/) were searched. In addition, the international databases OpenGrey (http://www.opengrey.eu/) and Russian Scientific Library, Library of dissertations (http://freereferats.ru/index.php?cat=91&page=14) were searched. Furthermore, the official websites of Russian government agencies were searched for official reports and overviews as well as epidemiological bulletins, using the same search terms in Russian.

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
