# Peer review of "Human Echinococcosis in the Russian Federation in the 21st Century: A Systematic Review"

_microorganisms, 2025, doi:10.3390/microorganisms13051122_

Round 1

Reviewer 1 Report

Comments and Suggestions for Authors

This paper presents a systematic review of the epidemiological characteristics of human echinococcosis (Cystic Echinococcosis, CE, and Alveolar Echinococcosis, AE) in the Russian Federation from 2000 to 2021, filling a gap in the data within this field. The research design is reasonable, with diverse data sources, including published literature, grey literature, and official reports. The analytical methods are rigorous. The results clearly demonstrate the decline in the incidence of CE, the geographical distribution differences, and the risk factors, which are of great reference value for the formulation of public health policies.

  1. It is necessary to supplement the specific content of the PRISMA flowchart (Figure S1) and clarify the number of excluded studies and the reasons (for example, define precisely what is meant by "duplicate data").
  2. Contradiction in Time Scope: The abstract mentions that the research period is from 2000 to 2021, while the search scope in the Materials and Methods section is from 2001 to 2023. The expression needs to be unified.
  3. The results indicate that CE has a higher incidence in rural areas, but some regions (such as the Far Eastern Federal District) do not show such differences. The reasons, such as climate and population migration, need to be further explored.
  4. The mortality rate of AE is significantly higher than that of CE, yet there is no discussion in combination with clinical characteristics (such as diagnostic delay and treatment difficulty).
  5. The paper mentions that the prevention and control policies after 2014 may have contributed to the decline in the incidence of CE, but there is a lack of data support on the specific implementation effects (such as the deworming coverage rate and the implementation rate of slaughter supervision).
  6. Supplement Risk Factor Analysis: The paper mentions that "contact with dogs" is a major risk factor for CE, but it has not been quantified (such as the proportion of dog-owning households and the frequency of deworming).

This paper provides important data support for the epidemiological study of echinococcosis in the Russian Federation. The methods are rigorous, and the conclusions are reasonable. However, it needs to be further improved in terms of data transparency, chart optimization, and the depth of discussion. It is recommended to accept the paper for publication after minor revisions.

Author Response

Comments 1: It is necessary to supplement the specific content of the PRISMA flowchart (Figure S1) and clarify the number of excluded studies and the reasons (for example, define precisely what is meant by "duplicate data").

Response 1: Done, diagram content has been added.

Comments 2: Contradiction in Time Scope: The abstract mentions that the research period is from 2000 to 2021, while the search scope in the Materials and Methods section is from 2001 to 2023. The expression needs to be unified.

Response 2: Additional explanation has been added in section 2.1 Search strategy by line 88 to line 90

Comments 3: The results indicate that CE has a higher incidence in rural areas, but some regions (such as the Far Eastern Federal District) do not show such differences. The reasons, such as climate and population migration, need to be further explored.

Response 3: Done changes have been made – by line 380 to line 387

Comments 4: The mortality rate of AE is significantly higher than that of CE, yet there is no discussion in combination with clinical characteristics (such as diagnostic delay and treatment difficulty).

Response 4: Changes have been made – by line 409 to line 412. We did not collect data on clinical characteristics of the infection.

Comments 5: The paper mentions that the prevention and control policies after 2014 may have contributed to the decline in the incidence of CE, but there is a lack of data support on the specific implementation effects (such as the deworming coverage rate and the implementation rate of slaughter supervision).

Response 5: Unfortunately, we did not find more detailed information

Comments 6: Supplement Risk Factor Analysis: The paper mentions that "contact with dogs" is a major risk factor for CE, but it has not been quantified (such as the proportion of dog-owning households and the frequency of deworming).

Response 6: Changes have been made – by line 247 to line 255

Reviewer 2 Report

Comments and Suggestions for Authors

The article summarizes and analyses the data on cystic and alveolar echinococcosis in the Russian Federation from 2000 to 2021. Publications, doctoral theses and official reports were used as sources.

The introduction introduces the topic. However, the working hypothesis could be made clearer. It is not immediately clear why the period between 2000 and 2021 should be specifically considered. The structure of the article is almost identical to the article on the incidence of taeniasis in the Russian Federation mentioned in source 11. In that article, the authors made a link between socio-economic development in the Russian Federation and the implementation of prevention programmes. A similar link should be made in the introduction to the article on echinococcosis.

Methods are clearly described and reproducible.

Results are clear and well documented. However, I find the structure of the results section, divided into "echinococcosis", "cystic echinococcosis" and "alveolar echinococcosis", somewhat confusing. In my opinion, the structure would be clearer if it were organised as indicated in the introduction, i.e. incidence, demographic and geographical distribution, risk factors and mortality. This would allow the influence of documented factors on outcomes, such as inadequate diagnosis, poor reporting discipline or inadequate prevention, to be more clearly highlighted.

The discussion raised the issue of data quality. This is a phenomenon that affects all data that enter official reporting systems. It relativises the significance of the results. In listing possible preventive measures, it is not clear whether these are to be implemented first or are already in place and whether they are effective or not. The discussion does not include a classification of the work within existing research. If this does not already exist, I think it should be mentioned.

I would classify this paper as a meta-analysis rather than a systematic review. The compilation and evaluation of the data is a valuable contribution to the description of the epidemiology of echinococcosis in the Russian Federation. However, I would structure the results section more clearly to make it easier to understand.

Check the hyperlink in citation 14. It does not seem to work. 

Author Response

Comments 1: The introduction introduces the topic. However, the working hypothesis could be made clearer. It is not immediately clear why the period between 2000 and 2021 should be specifically considered. The structure of the article is almost identical to the article on the incidence of taeniasis in the Russian Federation mentioned in source 11. In that article, the authors made a link between socio-economic development in the Russian Federation and the implementation of prevention programmes. A similar link should be made in the introduction to the article on echinococcosis.

Response 1: Changes have been made – by line 72 to line 81

Comments 2: Results are clear and well documented. However, I find the structure of the results section, divided into "echinococcosis", "cystic echinococcosis" and "alveolar echinococcosis", somewhat confusing. In my opinion, the structure would be clearer if it were organised as indicated in the introduction, i.e. incidence, demographic and geographical distribution, risk factors and mortality. This would allow the influence of documented factors on outcomes, such as inadequate diagnosis, poor reporting discipline or inadequate prevention, to be more clearly highlighted.

Response 2: As cases of cystic and alveolar echinococcosis were collectively registered until 2014, it was only possible to separate data related to cystic and alveolar echinococcosis from 2014. That conditioned the structure of the results section. We considered the proposed organization of the results, but as this would require each chapter to have three sub-units, we opted for this approach hoping that it would be more transparent.

Comments 3: The discussion raised the issue of data quality. This is a phenomenon that affects all data that enter official reporting systems. It relativises the significance of the results.

Response 3: We agree and that is why we emphasized it in the discussion. We also believe that even if the registration is incomplete, it is not selective and does not affect the relationships between the data obtained from individual subgroups.

Comments 4: In listing possible preventive measures, it is not clear whether these are to be implemented first or are already in place and whether they are effective or not.

Response 4: The above measures have been applied since 2014 (changes have been made – by line 418 to line 419). However, there is no specific data on the extent of their application. Therefore, we can only assume their effectiveness based on the decline in the prevalence of C.E.

Comments 5: The discussion does not include a classification of the work within existing research. If this does not already exist, I think it should be mentioned.

Response 5: In the third line of the discussion (line 305) specifies that this is a systematic review.

Comments 6: I would classify this paper as a meta-analysis rather than a systematic review.

Response 6: Our assessment was that the collected data were not adequate for a high-quality meta-analysis, so we decided to conduct a systematic review.

Comments 7: Check the hyperlink in citation 14. It does not seem to work.

Response 7: We checked and were able to open the site.